# Designing Highly Efficient Temperature Controller for Nanoparticles Hyperthermia

**DOI:** 10.3390/nano12193539

**Published:** 2022-10-10

**Authors:** Adeel Bashir, Sikandar Khan, Salem Bashmal, Naveed Iqbal, Sami Ullah, Liaqat Ali

**Affiliations:** 1Department of Electrical Engineering, COMSATS University, Islamabad 45550, Pakistan; 2Department of Mechanical Engineering, King Fahd University of Petroleum and Minerals, Dhahran 31261, Saudi Arabia; 3Interdisciplinary Research Center for Intelligent Manufacturing and Robotics, King Fahd University of Petroleum and Minerals, Dhahran 31261, Saudi Arabia; 4Department of Electrical Engineering, King Fahd University of Petroleum and Minerals, Dhahran 31261, Saudi Arabia; 5Center of Energy and Geo Processing, King Fahd University of Petroleum and Minerals, Dhahran 31261, Saudi Arabia; 6K. A. CARE Energy Research & Innovation Center (ERIC), King Fahd University of Petroleum and Minerals, Dhahran 31261, Saudi Arabia; 7College of Civil Engineering & Architecture, Zhejiang University, Hangzhou 310058, China

**Keywords:** temperature controller, nanoparticles hyperthermia, control system hyperthermia, PID controller, robust control, magnetic hyperthermia

## Abstract

This paper presents various control system design techniques for temperature control of Magnetic Fluid hyperthermia. The purpose of this research is to design a cost-effective, efficient, and practically implementable temperature controller for Magnetic Fluid hyperthermia, which is presently under research as a substitute to the radiation and chemotherapy treatment of cancer. The principle of this phenomenon centers on the greater sensitivity of tumor cells to changes in temperature in comparison to healthy cells. Once the nanoparticles reach the desired tissue, it can then be placed in a varying magnetic field to dissipate the heat locally by raising the temperature to 45 °C in order to kill cancerous cells. One of the challenging tasks is to maintain the temperature strictly at desired point i.e., 45 °C. Temperature controller for magnetic fluid hyperthermia provides the tight control of temperature in order to avoid folding of proteins and save the tissues around the cancerous tissue from getting destroyed. In contrast with most of the existing research on this topic, which are based on linear control strategies or their improved versions, the novelty of this research lies in applying nonlinear control technique like Sliding Mode Control (SMC) to accurately control the temperature at desired value. A comparison of the control techniques is presented in this paper, based on reliability, robustness, precision and the ability of the controller to handle the non-linearities that are faced during the treatment of cancer. SMC showed promising results in terms of settling time and rise time. Steady state error was also reduced to zero using this technique.

## 1. Introduction

Cancer is one of the most common and deadliest diseases with a high mortality rate [1]. There are different types of treatments that are employed to cure this disease such as surgery with chemotherapy, immunotherapy, targeted therapy, radiation therapy or hormone therapy. All the mentioned treatments bear their own limitations such as poor accessibility to tumor tissue [2]. For example, some tumors are inoperable due to their localization [3]. Moreover, the techniques mentioned above have a high degree of toxicity and are somewhat ineffective in many cases [4]. Hyperthermia has been utilized in combination with radiotherapy for many years, but posed the serious side effects of damaging healthy tissues. In order to overcome these limitations, scientists are constantly looking for alternative antitumor therapies [5,6]. The use of nanoparticles in the field of medical science is one of the emerging trends [7]. Magnetic fluid hyperthermia provides an efficient and effective solution to this issue [8], as this technique is based on the use of magnetic nanoparticles (mNPs) to induce local heat remotely, when an alternating magnetic field is applied [9]. This will increase only the temperature of the tissue harboring the tumor as shown in Figure 1, where mNPs are injected into the tumor of a mouse which is then placed inside an Alternating Magnetic Field (AMF). Magnetic nanoparticles have attracted the attention of researchers worldwide due to their diversity [10,11,12,13]. Most of the experiments performed with mNPs for the treatment of cancer are via direct injection of the particles into the tumor tissue [14].

One of the most significant factors that decides the effectiveness of this method is the ability of nanoparticles to be easily driven and collected in the organ of interest inside the body. Magnetic hyperthermia (MHT) is a promising approach for cancer therapy [16]. Gilchrist et al. [17] were the first scientists to use maghemite (γ-Fe_2_O_3_) nanoparticles for hyperthermia in the treatment of cancer. The nanoparticles used for magnetic fluid hyperthermia should respond to physical-chemical properties such as aggregation state, production of heat dose, conversion of heat from magnetic energy and surface chemistry [18]. Studies have also shown that ferrite nanoparticles are one of the candidates for the mNPs [19]. Similarly, superparamagnetic iron oxide nanoparticles (SPIONs) are also a good option due to their biocompatibility, excellent response to external magnetic field, non-toxic nature and ease of production [20].

The process of killing tumor cells through hyperthermia can be explained by the flow chart shown in Figure 2 [21], which illustrates the process of hyperthermia in detail. It can be inferred from the chart that hyperthermia will initiate two processes; one is an increased metabolic rate, which in turn increases the generation of reactive oxygen species and second is protein damage through oxidation, aggregation and denaturation. Hyperthermia will lead to nuclear protein damage, which will then inhibit the DNA repair mechanism; similarly, it will cause cell proliferation via G1 cell cycle arrest or mitotic catastrophe. Moreover, hyperthermia can also lead to membrane damage, which will alter the transport function, signaling mechanism or receptor function of the cell. All the cell changes explained above will lead to death of the tumor cell. In tissues, heat is generated by metabolism and blood perfusion, and the heat generated during metabolic processes, such as growth and energy production of the living system, is defined as metabolic heat. The effect of the temperature increase during metabolism of the material can be reduced by lowering the frequency of the alternating magnetic field, hence the net heat will remain the same. There is a need for a simple analytical solution to evaluate the effect of parameters such as metabolic heat generation during hyperthermia.

In 2003, initial clinical trials of magnetic fluid hyperthermia using alternating magnetic field (AMF) exposure of high frequency were conducted in fourteen glioblastoma multiform subjects [22]. One promising feature about this technique is that it can be used in regions that are difficult to access owing to the intravenous administration route of mNPs [23]. In a recent study, it was proved that mild magnetic nanoparticles have activated immunity against liver cancer [24].

In cases of overheating of the human body above 43 °C, serious violations occur in life-support systems and heat stroke develops. Further increases in temperature cause an irreversible violation of the structure and function of protein molecules leading to tissue death. In this regard, attention is drawn towards killing of cancerous tissues using local hyperthermia; however, the ideal limits of heating the tissues are still under discussion. Many researchers have defined the anticancer activity of 41–43 °C (minor hyperthermia) [25,26,27,28,29] or ≥50 °C (thermal ablation) [30,31] treatments. However, there are only few explorations, which include direct comparison of the heating doses’ effectiveness during magnetic hyperthermia, 43 °C with 50 °C and similarly, 50 °C with 60 to 70 °C. Temperature must be controlled tightly in local hyperthermia so that only the tumor tissues are killed, and the healthy tissues are unaffected. An efficient controller is required to maintain the temperature at the desired value and reduce the input heat supply at the appropriate time. To this end, various close-loop techniques have been proposed [32,33,34] but they mainly focus on MRI-guided HIFU therapy.

Magnetic Fluid Hyperthermia showed positive results in recent clinical experiments. Therefore, it is likely to empower the existing approaches in the treatment of cancer [35,36]. To restrict the heat diffusion, the accurate measurement of the temperature and controlling of the magnetization current are important. The principle of this phenomenon is the fact that tumors cells are more sensitive to temperature changes than local cells. The local cells are those that are healthy or free from any sort of mutation. Moreover, clinical studies have proven that the walls of mutated cells are more porous than normal cells as shown in Figure 3, allowing the easy deposition of nanoparticles on tumors cells. Once the nanoparticles reach the desired tissue, they can be placed under a varying magnetic field to dissipate the heat locally, and the temperature raised to 45 °C to kill cancerous cells. The set point of the temperature controller, i.e., 45 °C is completely safe for normal tissue. One of the challenging tasks is maintaining the temperature strictly at the desired point i.e., 45 °C. A temperature controller for magnetic fluid hyperthermia provides the tight temperature control needed in order to avoid the folding of proteins and to prevent destruction of the tissues around the cancerous tissue. The purpose of the current research work is to propose an efficient temperature controller for magnetic fluid hyperthermia. In contrast with most of the existing research on nanoparticle hyperthermia, which are based on linear control strategies or their improved versions, the novelty of this research lies in applying robust control technique, i.e., Sliding Mode Control (SMC) to accurately control the temperature at the desired value.

## 2. Delivery of Nanoparticles to Area of Interest

There are two ways in which nanoparticles can be delivered to the cancerous tissue, i.e., direct injection of nanoparticles into the tissue [37,38,39,40] and intravenous injection. Direct injection of nanoparticles into the tissue was used in past but some drawbacks and limitations were found with this method. Firstly, it was almost impossible to determine whether all the particles injected had reached the affected tissue or not. Secondly, if the tumor lies under a sensitive organ, then it might be difficult to reach the affected tissues. The injection of particles directly into the tissue came with other problems including potentially missing an area of the tumor thus allowing cancerous cells to regrow after some time. Currently, doctors are using a new way of delivering nanoparticles to the cancerous tissue. Intravenous injection is a technique that has been used for centuries for the delivery of drugs and is now used for magnetic fluid hyperthermia [41,42,43,44].

The laboratory experiments show that the accumulation of nanoparticles is 16% more in the case of intravenous injection than in direct injection of the particles into the tumor. However, one thing that troubled the doctors administering these particles intravenously was the potential for iron toxicity. Scientists sought out this issue in a better way; they wrapped the particles in a biocompatible shell to alleviate its toxicity. As shown in Figure 4, Yogita et al. [45] discussed the biocompatibility of the super paramagnetic core-shell nanoparticles for possibility of their future use in magnetic fluid hyperthermia. Similarly, Priti Ghutepatil et al. [46] synthesized nanoparticles from manganese ferrite with a biocompatible shell composed of polyvinylpyrrolidone (PVP). Another very promising research work was conducted by Reza Eivazzadeh-Keihan et al. [47] who used carboxymethyl cellulose (CMC) and epichlorohydrin (ECH) for the production of nanoparticles to be used for the hyperthermia application. After the outbreak of COVID-19, researchers began looking for antimicrobial biocompatible nanoparticles that can be used as antibacterial and antifungal agents. Diksha Pathania et al. [48] conducted a similar study in which they synthesized essential-oil-mediated biocompatible magnesium nanoparticles, which bear antifungal and antibacterial properties. These particles can also be used for magnetic fluid hyperthermia, which will reduce the risk of obtaining bacterial or fungal infections at the tumor site.

The particles inside the shell will never come in to direct contact with the body. With this, toxicity is avoided. There are different types of polymers that are used to coat the nanoparticles, some are given in Table 1 below [49]. During this research, commercially available spherical SPIONs (superparamagnetic iron oxide) from Skyspring Nanomaterials Inc., Houston, TX, USA (3327NG.) nanoparticles are used, whose average size is 10–15 nm.

## 3. Feasible Temperature Measuring Techniques for Magnetic Fluid Hyperthermia

The first and most important task for magnetic fluid hyperthermia technique is to find an efficient way for measuring the temperature of the tumor site. Although there are many techniques to measure temperature, for this specific task, it is very difficult to find a technique that suits the requirements. The temperature-measuring technique must have the following six properties:Can measure temperature at depth.Have high accuracy.Non-invasive.Can be used practically with ease.Should not be bulky.Should be economical.

Different types of measuring instruments/techniques are available today for the measurement of temperature. For instance, tiny thermometers are used around the globe for temperature measurement. They have almost 95% accuracy, are not bulky and are economical. However, one of the major reasons it cannot be used in magnetic fluid hyperthermia is that it cannot measure temperature at depth and is invasive. Similarly, ultrasound methods are also used for temperature measurement. They are noninvasive, portable and are not bulky and can measure temperature at depth. However, the temperature sensitivity of this technique is much lower and due to this reason, it cannot be used for the desired treatment. Another temperature-measuring technique, Magnetic Resonance Imaging (MRI) is very accurate [58,59,60] as compared to other methods described above; however, the major disadvantage of this technique is that it is bulky. The fiber-optic temperature measurement technique has also attracted the attention of researchers in recent times; however, one of the major drawbacks of this technique is that the accuracy of sensors vary widely. Similarly, the development of the sensor is highly complex and some are very expensive [61]. Thermal-imaging infrared cameras can be imagined as a potential candidate for magnetic fluid hyperthermia but it has the major drawback of inaccurate measurement. This is due to the fact that measurements are hindered by differing emissivity and reflections from surfaces. Moreover, it is a costly technique and requires high initial investment. One of the most recent techniques used for temperature measurement is the photoacoustic effect. In this technique, the temperature is measured by using the following formula:(1)$$T=\frac{P}{P\left(T{}^{\circ}\right)}\left(\frac{A}{B}+T{}^{\circ}\right)-\frac{A}{B}$$
where, T = measured temperature, P = amplitude of acoustic wave, A and B = constants determined by material, P (T°) = known reference point of temperature, T° = initial temperature.

This technique is non-invasive, economical and practically useful for magnetic fluid hyperthermia.

## 4. System Modeling

After selecting the appropriate technique for temperature measurement, the next step is to design the controller for the magnetic fluid hyperthermia. The controller acts as the brain of this system. However, before going into the detail of the controller, it is necessary to understand the system as a whole and construct a mathematical model of the system. A block diagram of the system is shown in Figure 5.
Figure 5Block diagram of the system.
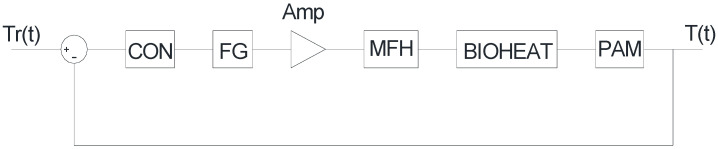
where,

CON = Controller

FG = Function generator

AMP = Amplifier

MFH = Magnetic Ferro fluid heating

BIOHEAT = Bio heating effect

PAM = Photo Acoustic Measurement

Tr(t) = Reference temperature

T(t) = Measured temperature

The controller is the brain of the system. It will control the temperature of the system. The input of the controller is the error. The controller will decide the variation of magnetic field in order to achieve the threshold of the temperature and then stop the field variation so that temperature will not exceed the preset value of the temperature. In this way, the controller will save the adjacent tissues from destruction. The function generator used in this system is not a complex one. It simply obtains the power value from the control unit and produces a physical radio frequency to feed the amplifier. The amplifier used is the power amplifier with unity gain. These two can be merged into one block with power Gain “Gᵢ”.

Magnetic Ferro fluid heating (MFH) describes the phenomena behind the heating of the nanoparticles under a magnetic field. The term Ferro fluid refers to a liquid that is strongly magnetized in the presence of the magnetic field. As nanoparticles are injected within a mixture of liquid, it will become magnetized in the presence of the magnetic field. When placed in an alternating magnetic field, mNPs will undergo Brownian and Neel relaxation processes, resulting in energy absorption from the magnetic field and this energy is responsible for heating the ferrofluid. The volumetric power deposition is given by U in Equation (2). The heating of the fluid is governed by the first law of thermodynamics. The mathematical equation used for the Ferro fluid is given below [62],
(2)$$U=\frac{\left\{\left(\frac{\mu }{2}\right){\;\times \;H}_{0}^{2}{\;\times \;X}_{0\;}{\times \;\omega }^{2}{\;\times \;T}_{a}\right\}}{\{1+{\left({\omega \;\times \;T}_{a}\right)}^{2}\}\;}$$
where,

U = Volumetric power deposition

μ = Permeability of free space (F/m)

$H_{0}$ = Magnetic field strength

$X_{0}$ = Magnetic susceptibility

ω = Radian frequency

$T_{a}$= Effective relaxation time

The most complex part of the system pertains to Bioheat. It is basically a mathematical model of the tissue where the tumor is located. The basic characteristic of all tissues remains the same so one generic model can be used. The dependence of this will be on the following human parameters:Nanoparticle heatingBlood convectionHeat diffusion

Considering all the parameters mentioned above, the following equation can be used for modeling the Bioheat [63],
(3)$$\rho \left(r\right)c\left(r\right)\left(\frac{\partial T\left(r,T\right)}{\partial t}\right)=\nabla K\left(r\right)*\nabla T\left(r.t\right)+U\left(r,t\right)+{\;c}_{b\;}\omega \left(T_{a}-T\left(r,t\right)\right)\;$$
where,

$\rho \left(r\right)$ = mass density of nanoparticle-laden tissue

c(r) = specific heat of nanoparticle-laden tissue

K(r) = thermal conductivity of laden tissue

$c_{b}$ = specific heat of blood

ω = local perfusion rate of blood

$T_{a}$ = temperature of the blood

$T\left(r,T\right)$ = temperature evaluation inside patient

$U\left(r,t\right)=$ time rate of electromagnetic energy deposited per unit volume

Equation (3) is simplified in order to obtain an appropriate analytical solution of the system. If we consider that there is a sharp roll-off of the temperature in the tissues laden with nanoparticles, then the equation simplifies to [64,65,66],
(4)$$\rho c\left(\frac{\partial T\left(t\right)}{\partial t}\right)=-R\left(T\left(t\right)-{\;T}_{a\;}\right)+\;U\left(t\right)+{\;c}_{b\;}\omega \left(T_{a}-T\left(t\right)\right)$$
where, R = characteristic time constant of the tissue–nanoparticles system.

With all blocks modeled and after taking their Laplace, the entire system can be represented with a transfer function as given below,
(5)$$Q_{s}=\frac{\frac{G_{1}G_{2}}{\rho c}}{s+\;\beta /\rho c}e^{-\mathrm{sL}}$$
where,

β = Local blood perfusion constant

$G_{1}$ = Power amplifier gain

$G_{2}$ = Gain for local thermal diffusion

## 5. Controller Design: Simulation and Results

In total, three different types of control techniques are applied on the above-modeled system, their simulation and results are explained below, and their comparison is given in the next section.

### 5.1. PID Controller

In nanoparticle hyperthermia temperature control, all three combinations of the PID controller are applied. MATLAB software is used for modeling, as used by various studies in the literature [67,68,69]. MATLAB Simulink (R2021a) is used to define the system transfer function as functional blocks and then a PID Controller block is linked to the input of the plant as shown in Figure 6.

The value of the preset temperature was set to 45 °C. After adjusting the parameters and tuning the PID parameters, a step response of the system was obtained from the scope. Step response of the plant using PID, PI and PD controller is shown in Figure 7.

It can be seen from Figure 7 that only the PI controller will reach the desired temperature of 45 °C, whereas PID reaches 44 °C and PD reaches 43 °C. So, only PI controller will be further discussed.

There are two parameters of this controller that are necessary to discuss here, one is the rise time of the system and the other is the settling time. The system should reach and settle on the desired value as fast as possible with no overshoot in the temperature. The overshooting of the temperature is lethal.

The PI was fine-tuned to attain the desired results. Manual tuning methods were used to tune the parameters of the Kp and Ki in order to achieve the desired response. The system should reach and settle on the desired value as fast as possible with no overshoot in the temperature. The response of the system is shown in Figure 8.

It can be seen from the response that the rise time of the system is 100 s and the settling time of the system is 500 s.

### 5.2. Pole Placement Technique

The main idea behind this technique is to transform the state-space representation of the system in to the standard form as given below,
(6)x=Ax+Bu
(7)y= Cx+ Du
where,

A = State matrix

B = Input Matrix

C = Output Matrix

D = Feed Through Matrix

u = Input

y = Output

x = State Vector

The roots of the system are calculated and then full state feedback is used to form the general equation given below for the K matrix. This K matrix is used for controlling purposes of the plant.
(8)u=−Kx
where, K = Feedback matrix.

The Simulink model (R2021a) of the complete nanoparticles hyperthermia system with pole placement controller design is shown in Figure 9.

In the above Simulink model, it is shown that the error is calculated from preset value of the temperature and from the actual temperature value measured from the cancerous tissue laden with nanoparticles. The measured error value is then fed to the pole placement controller block. Depending on the error, the controller generates the output and the tissue containing the tumor is modeled in the block on which magnetic field is applied.

After adjusting the parameters and tuning the K parameter, a step response of the system was obtained from the scope and the result is shown in Figure 10.

Manual tuning methods were used to tune the parameters of the K matrix. It can be seen from the response that the rise time of the system is 8 s and settling time of the system is 25 s.

Although the obtained response is acceptable in various aspects, another issue arises regarding the presence of a steady-state error in the response. It can be seen from the graph that the response never reaches 45 °C, it reaches 44.2 °C and remains there with a steady-state error of 0.8 °C. However, the error is small and can be ignored from a mathematical point of view but it may leave some of the tumor cells alive, which can trigger the disease again in future. Another important issue that needs to be highlighted here is that the rise time of the system is too small, i.e., 8 s. This might result in thermal shock to the tissue.

### 5.3. Sliding Mode Control of Magnetic Fluid Hyperthermia

In order to design the sliding mode control for nanoparticles hyperthermia, the system was first converted into state space. After obtaining the state-space representation, there were two major steps in designing the control, i.e., defining the sliding surface and derivation of control input “U”.

Sliding surface was defined as given below,
(9)$$S={\left(\frac{d}{\mathrm{dt}}+\lambda \right)}^{n-1}e$$
(10)$$e=T_{d}-T_{m}$$
where,

S = Sliding surface

e = error

T_d_ = Desired temperature value

T_m_ = Measured temperature value

λ = Tuning parameter

From Figure 11, it is revealed that the error is calculated from the preset temperature value and from the measured temperature value from the cancerous tissue laden with nanoparticles. This error is then fed in to the sliding-mode controller block. Depending on the error, the controller generates the output that controls magnetic Ferro fluid heating. Bio heat, i.e., the tissue containing the tumor was modeled in this block to which a magnetic field is applied. A step response of the system was obtained from the scope and the result is shown in Figure 12. It can be seen from the graph that there is a significant improvement in the settling time and rise time of the system.

The parameters recorded are rise time of 170 s and settling time of 380 s. As shown in Figure 12, the rise time, i.e., 170 s is sufficient to avoid thermal shock to the tissue and the system settles down quickly after obtaining the desired value. There was no overshoot in the system so the tissue around the tumor region is safe from the heat diffusion from the site. Similarly, the steady-state error was also eliminated, as was present in the case of the pole placement technique. The system smoothly arrives at the desired value of temperature and remains there.

## 6. Conclusions

After obtaining the response of the system via different control techniques, a comparison between them can be made. A summary of the parameters obtained from the various techniques is given in Table 2 and Figure 13.

PID controller of the function PI is used to achieve the results. The Kd gain of the PID controller was set to zero for this purpose. This control technique provides good results with no steady-state error; however, the settling time was large, which might result in heat diffusion to the surrounding tissues. The PID was fine-tuned to decrease the settling time. However, by reducing the settling time, the rise time also decreased, which will result in thermal shock to tumor-affected tissues and alleged tissues. As the rise time decreases, the temperature of the tissue laden with nanoparticles rapidly increases leading to the thermal shock situation.

Moreover, the pole placement method also introduced the steady-state error. Although the error is small in terms of mathematical figures, with regard to treating cancer, it is significant. The last technique used was a nonlinear control technique, i.e., sliding mode control. Sliding mode control was used to control the temperature of nanoparticles in the magnetic fluid hyperthermia approach. This technique showed promising results in terms of settling time and rise time. The steady-state error was also reduced to zero using this technique. Moreover, the rise time of the system remained within safety limits in order to avoid thermal shock. The settling time was also within safety limits in order to prevent heat diffusion to the surrounding tissues. Finally, it can be concluded that among the three techniques implemented, the sliding mode control is best suited for nanoparticle-based hyperthermia with respect to all three parameters.

## 7. Future Work

Magnetic fluid hyperthermia presents a challenging field for control engineers. This research opens a new dimension of nonlinear control techniques for use in linear systems. In future, more efficient nonlinear techniques may need to be implemented in order to achieve more promising results. Similarly, another area for future researchers to investigate concerns the mathematical modelling of this system, since the modelled uncertainties and the sensor noise are neglected. In future, this system may be modelled with uncertainties and sensor noise rendering it a MISO system. In doing so, robustness of the controller may be introduced. Robust controller design for magnetic fluid hyperthermia will increase its reliability and performance.

## Figures and Tables

**Figure 1 nanomaterials-12-03539-f001:**
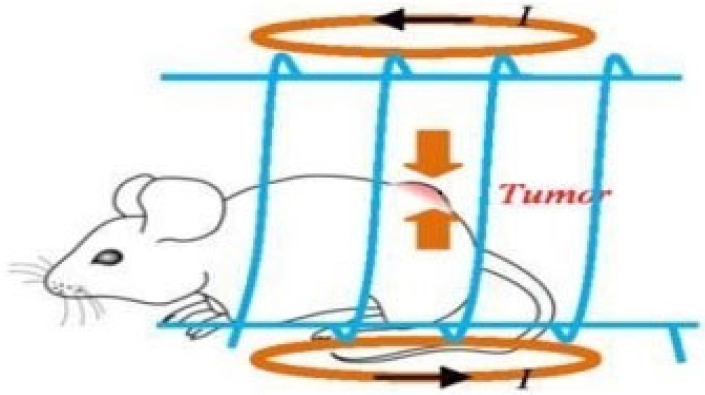
Tumor heating in magnetic field [15].

**Figure 2 nanomaterials-12-03539-f002:**
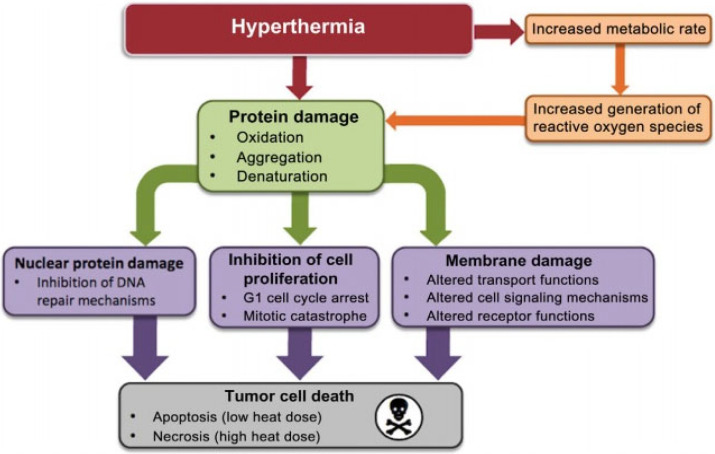
The induced cell changes due to hyperthermia [20].

**Figure 3 nanomaterials-12-03539-f003:**
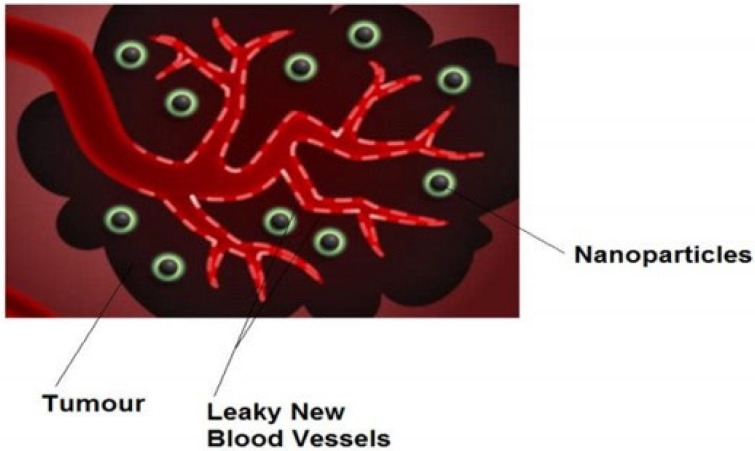
Accumulations of nanoparticles in tumor.

**Figure 4 nanomaterials-12-03539-f004:**
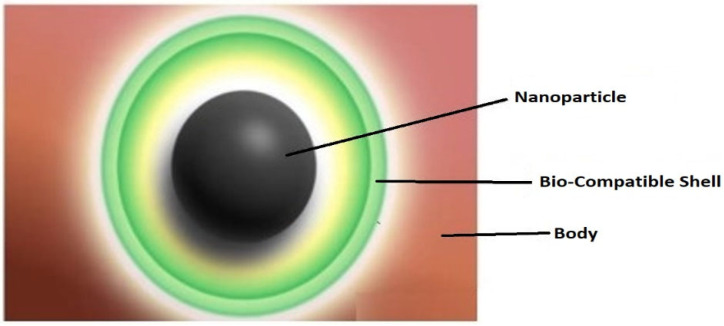
Biocompatible shell for nanoparticles.

**Figure 6 nanomaterials-12-03539-f006:**
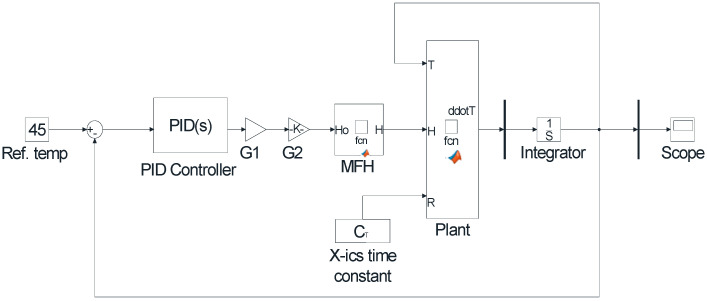
Simulink model of the plant with PID controller.

**Figure 7 nanomaterials-12-03539-f007:**
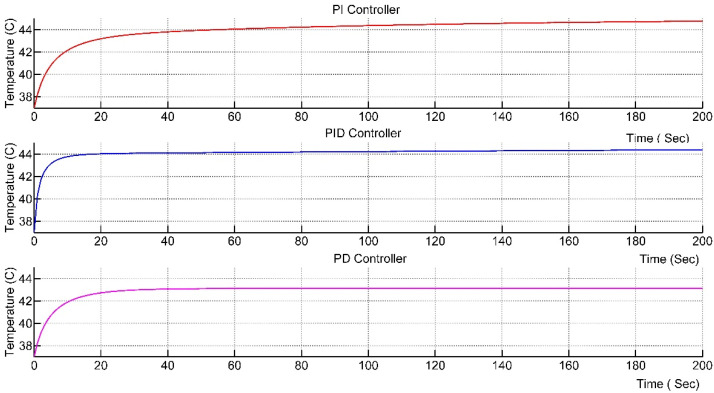
Step Response of PI, PID and PD controllers.

**Figure 8 nanomaterials-12-03539-f008:**
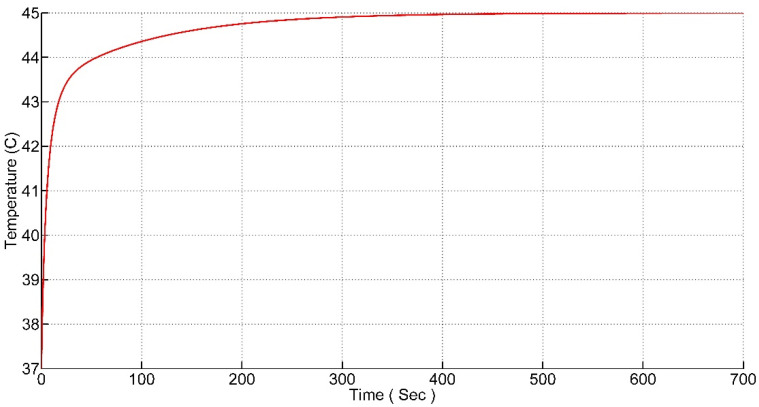
Step Response with PI controller.

**Figure 9 nanomaterials-12-03539-f009:**
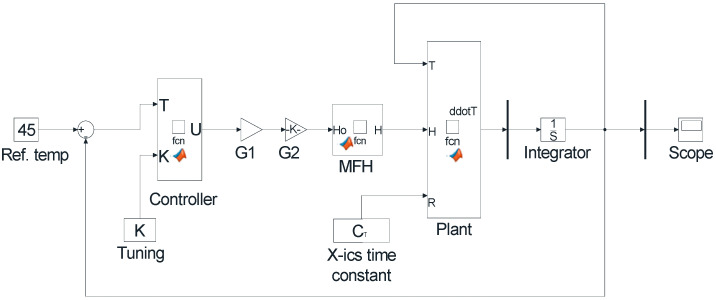
Simulink model with pole placement controller.

**Figure 10 nanomaterials-12-03539-f010:**
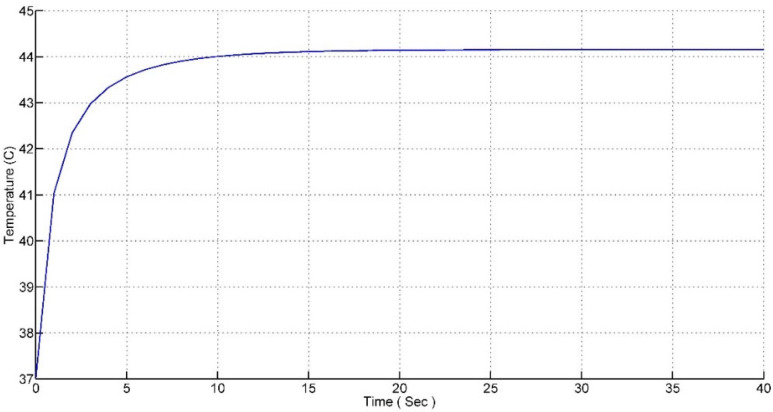
Step response of the pole placement controller.

**Figure 11 nanomaterials-12-03539-f011:**
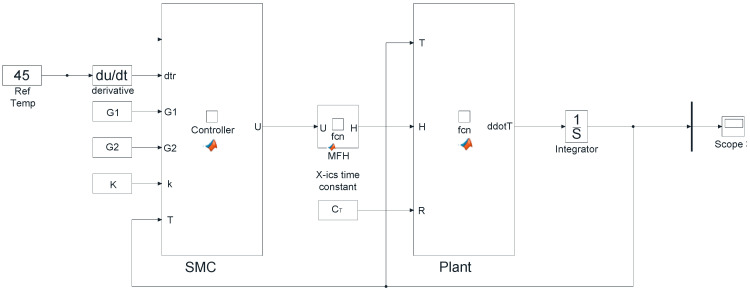
Simulink model of the plant with SMC controller.

**Figure 12 nanomaterials-12-03539-f012:**
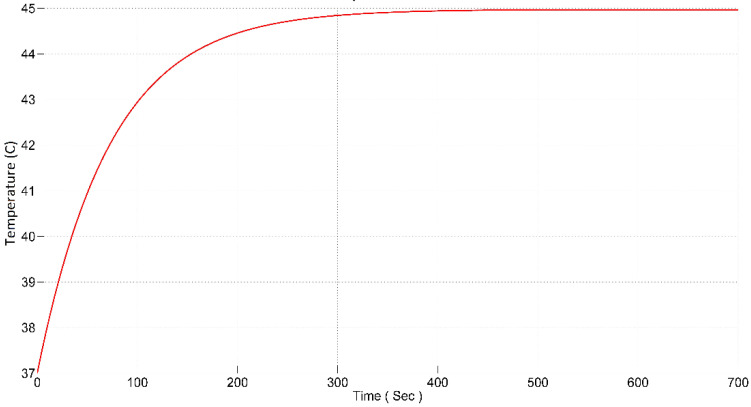
Step response of the plant with SMC controller.

**Figure 13 nanomaterials-12-03539-f013:**
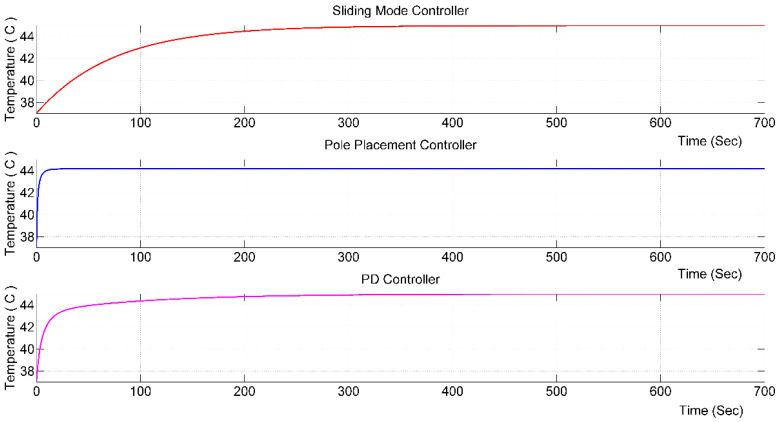
Combined step responses of all controllers.

**Table 1 nanomaterials-12-03539-t001:** Different materials used for coating of nanoparticles.

No.	Material	Reference
1	Gelatin	[50]
2	Dextran	[51]
3	Polyvinyl alcohol	[52]
4	Polyethylene glycol	[53]
5	Chitosan	[54]
6	Polyacrylic acid	[55]
7	Polyvinylpyrrolidone	[56]
8	Poly(D, L-lactide)	[57]

**Table 2 nanomaterials-12-03539-t002:** Comparison of the control techniques.

No.	Control Technique	Rise Time (s)	Settling Time (s)	Steady State Error (%)
1	Sliding Mode Control	170	380	0
2	Pole Placement Control Design	8	25	2
3	PI Controller	100	500	0

## Data Availability

The data are available by contacting the corresponding author.

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
