# Peer review of "Designing Highly Efficient Temperature Controller for Nanoparticles Hyperthermia"

_nanomaterials, 2022, doi:10.3390/nano12193539_

Round 1

Reviewer 1 Report

The manuscript may be of interest to readers, but it is quite "twisted" and thus difficult to follow. The information is not presented in a fluent manner, an aspect that should be remedied by the authors.

The novelty and relevance of the study do not emerge from the manuscript. In fact, it is not clear whether the study is original, or whether data from the literature is being discussed.

In addition, it is not clear if the mathematical model used to model and simulate the hyperthermia methods used in cancer treatment are validated, or if the study is incipient and requires further investigation. If the study is validated and can be practically applied to improve treatment methods, this aspect must be clearly mentioned in the manuscript. If, on the other hand, the study is incipient and requires deepening, the authors are asked to complete the manuscript with this information.

Observations and comments:

The authors are asked to explain what the term "State-of-the-art" in the title refers to. Wasn't the term "Proof-of-concept" more appropriate? In my opinion, "State-of-the art" fits better with a literature review type study, not with an original study. Please revise and/or change the term.

The authors are asked to clearly mention which figures are original and which are taken from the literature. Being an original study, my opinion is that it should not contain figures from the literature. If a comparison with a figure from the literature is desired, it is enough to refer to it in the text and cite it accordingly, and readers will consult it if needed.

Figures with modeling/simulation results are not uniformly represented. The authors should represent them in a uniform manner, so that they have the same dimensions on the axes and the same size for characters and numbers. That would be easier to follow.

The bibliographic references in Table 1 are quite old. The authors are requested to replace these references with more recent ones. Haven't there been any studies published in the meantime regarding coating materials for magnetic particles to ensure biocompatibility?

There are several errors in writing and expression, so the authors must carefully revise the entire manuscript.

Lines 114 -118:  Please explain the term “local cell”.

The term “the walls of these tissues” refers to “the walls of these cells”?

Line 191: the number of the figure is 5 not 4 as written in the text. Please revise

Line 263: it is “PI controller” or “PID controller”? Please revise

 To conclude, my opinion is that in its current form, the manuscript cannot be recommended for publication. Major revision and resubmission is necessary

Author Response

The following represent point-by-point answers to the reviewers’ comments. Appropriate revisions are made in the revised manuscript, as explained hereunder. The changes are highlighted in Turquoise.

Comments by Reviewer 1:

  • The authors are asked to explain what the term "State-of-the-art" in the title refers to. Wasn't the term "Proof-of-concept" more appropriate? In my opinion, "State-of-the art" fits better with a literature review type study, not with an original study. Please revise and/or change the term.

Answer: The authors are thankful to the reviewer for the valuable comment. As suggested, in the revised version of the manuscript, the term "state of the Art" has been replaced by “Designing Highly Efficient”. The updated title of the manuscript is now “Designing Highly Efficient Temperature Controller for Nanoparticles Hyperthermia”.

  • The authors are asked to clearly mention which figures are original and which are taken from the literature. Being an original study, my opinion is that it should not contain figures from the literature. If a comparison with a figure from the literature is desired, it is enough to refer to it in the text and cite it accordingly, and readers will consult it if needed.

Answer: The authors are thankful to the reviewer for the valuable comment. Only two figures (Figures 1 & 2) are taken from the literature, all other figures are original. As suggested, in the revised version of the manuscript, the figures taken from the literature are properly cited.

  • Figures with modeling/simulation results are not uniformly represented. The authors should represent them in a uniform manner, so that they have the same dimensions on the axes and the same size for characters and numbers. That would be easier to follow.

Answer: The authors are thankful to the reviewer for the valuable comment. As suggested, in the revised version of the manuscript, all the figures are revised in order have a uniform representation.

  • The bibliographic references in Table 1 are quite old. The authors are requested to replace these references with more recent ones. Haven't there been any studies published in the meantime regarding coating materials for magnetic particles to ensure biocompatibility?

Answer: The authors are thankful to the reviewer for the valuable comment. As suggested, in the revised version of the manuscript, the bibliographic references have been updated.

  • There are several errors in writing and expression, so the authors must carefully revise the entire manuscript.

Lines 114 -118:  Please explain the term “local cell”.

The term “the walls of these tissues” refers to “the walls of these cells”?

Line 191: the number of the figure is 5 not 4 as written in the text. Please revise

Line 263: it is “PI controller” or “PID controller”? Please revise

Answer: The authors are thankful to the reviewer for the valuable comment. As suggested, in the revised version of the manuscript, all corrections have been made. The manuscript has been read carefully for any possible typos and language mistakes. All the mistakes have been removed and the changes are highlighted in Turquoise.

Finally, the authors wish to thank the reviewer for his constructive remarks, which are well-taken and implemented to improve the clarity and quality of the manuscript.

Reviewer 2 Report

This research studies an efficient temperature controller for the magnetic  hyperthermia in order to avoid unnecessary heating to normal tissues using three different types of control techniques such as PID controller, Pole Placement Technique and Sliding Mode Control. There are major flaws in interpretation of results. The abstract should state briefly the purpose of the research, the principal results and major conclusions. The heating response of ferrofluid strongly depends on the size of magnetic nanoparticles and the amplitude field and frequency. In this study, the heating response is governed by the first law of thermodynamics. However, the heating response is generated by three different mechanisms as a function of the size: (a) Néel relaxation mechanism which describes the fluctuation of the magnetization through an energy barrier (b) Brown relaxation mechanism which is related to the rotation of the entire nanoparticle in the fluid and (c) hysteresis losses often modeled by the coherent rotation of the magnetization in magnetic nanoparticles with significant magnetic anisotropy. The authors must include the type (magnetite, maghemite e.t.c) and the size of magnetic nanoparticles for each model (see Ref. Iglesias, C. A. M., et al. "Magnetic nanoparticles hyperthermia in a non-adiabatic and radiating process." Scientific Reports 11.1 (2021): 1-13.). Moreover, the symbols in many equations have not described in the manuscript. The quality of the figures not suitable for publication. All figure captions have to be described with great detail in the manuscript.

Author Response

The following represent point-by-point answers to the reviewers’ comments. Appropriate revisions are made in the revised manuscript, as explained hereunder. The changes are highlighted in Yellow.

Comments by Reviewer 2:

  • The abstract should state briefly the purpose of the research, the principal results and major conclusions.

Answer: The authors are thankful to the reviewer for the valuable comment. As suggested, in the revised version of the manuscript, the abstract has been carefully revised.

  • The heating response of ferrofluid strongly depends on the size of magnetic nanoparticles and the amplitude field and frequency. In this study, the heating response is governed by the first law of thermodynamics. However, the heating response is generated by three different mechanisms as a function of the size: (a) Néel relaxation mechanism which describes the fluctuation of the magnetization through an energy barrier (b) Brown relaxation mechanism which is related to the rotation of the entire nanoparticle in the fluid and (c) hysteresis losses often modeled by the coherent rotation of the magnetization in magnetic nanoparticles with significant magnetic anisotropy. The authors must include the type (magnetite, maghemite e.t.c) and the size of magnetic nanoparticles for each model (see Ref. Iglesias, C. A. M., et al. "Magnetic nanoparticles hyperthermia in a non-adiabatic and radiating process." Scientific Reports 11.1 (2021): 1-13.).

Answer: The authors are thankful to the reviewer for the valuable comment. As suggested, in the revised version of the manuscript, this information is added in section 2, on page 5.

  • The symbols in many equations have not described in the manuscript.

Answer: The authors are thankful to the reviewer for the valuable comment. Missing symbols of the equations have been defined in the revised version of manuscript and are highlighted in yellow.

  • The quality of the figures not suitable for publication.

Answer: The authors are thankful to the reviewer for the valuable comment. As suggested, in the revised version of manuscript, the quality of the figures is improved.

  • All figure captions have to be described with great detail in the manuscript.

Answer: The authors are thankful to the reviewer for the valuable comment. As suggested, in the revised version of the manuscript, all figure captions of the figures, which were not explained previously have been described and highlighted.

Finally, the authors wish to thank the reviewer for his constructive remarks, which are well-taken and implemented to improve the clarity and quality of the manuscript.

Author Response

The following represent point-by-point answers to the reviewers’ comments. Appropriate revisions are made in the revised manuscript, as explained hereunder. The changes are highlighted in Green.

Comments by Reviewer 3:

  • Why choose 45℃ as the control temperature point? Whether 45℃ is completely safe for normal tissue?

Answer: The authors are thankful to the reviewer for the valuable comment. The set point of the temperature controller i.e. 45℃ is completely safe for normal tissue. Based on the published literature, hyperthermia treatment always have this set point. One of such study can been read from https://www.mdpi.com/2312-7481/5/4/67/htm#B19-magnetochemistry-05-00067. This information is added in section 1, on pages 3 & 4 of the revised manuscript.

  • How to reduce the effects of temperature rise on normal tissues during the metabolism of materials in vivo?

Answer: The authors are thankful to the reviewer for the valuable comment. In tissues, heat is generated by metabolism and blood perfusion, and the heat, that is, generated during metabolic processes, such as growth and energy production of the living system, is defined as metabolic heat. The effect of the temperature increase during metabolism of the material can be reduced by lowering the frequency of the alternating magnetic field, hence the net heat will remain same. There is a need for a simple analytical solution to evaluate the effect of parameters like metabolic heat generation during hyperthermia. This information is added in section 1, on page 3 of the revised manuscript.

  • Regarding the response time of tissue heat shock, is there clear literature supporting that 170 s can effectively avoid heat shock?

Answer: The authors are thankful to the reviewer for the valuable comment. The exact range for the time is not mentioned in published studies regrading heat shock of tissue. However, most of the published literature quotes that this time in in minutes. One of such study can be read from https://www.tandfonline.com/doi/pdf/10.1080/0265673031000119006.

Finally, the authors wish to thank the reviewer for his constructive remarks, which are well-taken and implemented to improve the clarity and quality of the manuscript.

Round 2

Reviewer 1 Report

The quality of the manuscript improved after the review.

The authors responded to the reviewers' comments and modified the manuscript according to the indications.

In this form, it can be recommended for publication.

Author Response

  • The quality of the manuscript improved after the review. The authors responded to the reviewers' comments and modified the manuscript according to the indications. In this form, it can be recommended for publication.

Answer: The authors are thankful to the reviewer for the valuable comments, which were well-taken and implemented in the revised manuscript and has improved the clarity and quality of the manuscript.

Reviewer 2 Report

The manuscript with the title "State-of-the-Art Temperature Controller for Nanoparticles Hyperthermia " is ready for publication. 

Author Response

  • The manuscript with the title "State-of-the-Art Temperature Controller for Nanoparticles Hyperthermia " is ready for publication.

Answer: The authors are thankful to the reviewer for the valuable comments, which were well-taken and implemented in the revised manuscript and has improved the clarity and quality of the manuscript.
